# Pharmacodynamic Model of the Dynamic Response of *Pseudomonas aeruginosa* Biofilms to Antibacterial Treatments

**DOI:** 10.3390/biomedicines11082316

**Published:** 2023-08-21

**Authors:** Swarnima Roychowdhury, Charles M. Roth

**Affiliations:** 1Department of Biomedical Engineering, Rutgers, The State University of New Jersey, Piscataway, NJ 08854, USA; sr1435@scarletmail.rutgers.edu; 2Department of Chemical and Biochemical Engineering, Rutgers, The State University of New Jersey, Piscataway, NJ 08854, USA

**Keywords:** pharmacodynamics, compartmental model, drug diffusion, biofilm

## Abstract

Accurate pharmacokinetic–pharmacodynamic (PK-PD) models of biofilm treatment could be used to guide formulation and administration strategies to better control bacterial lung infections. To this end, we developed a detailed pharmacodynamic model of *P. aeruginosa* treatment with the front-line antibiotics, tobramycin and colistin, and validated it on a detailed dataset of killing dynamics. A compartmental model structure was developed in which the key features are the diffusion of the drug through a boundary layer to the bacteria, concentration-dependent interactions with bacteria, and the passage of the bacteria through successive transit states before death. The number of transit states employed was greater for tobramycin, which is a ribosomal inhibitor, than for colistin, which disrupts bacterial membranes. For both drugs, the experimentally observed delay in the killing of bacteria following drug exposure was consistent with the sum of the diffusion time and the time for passage through the transit states. For each drug, the PD model with a single set of parameters described data across a ten-fold range of concentrations and for both continuous and transient exposure protocols, as well as for combined drug treatments. The ability to predict drug response over a range of administration protocols allows this PD model to be integrated with PK descriptions to describe in vivo antibiotic response dynamics and to predict drug delivery strategies for the improved control of bacterial lung infections.

## 1. Introduction

Bacterial biofilms contain cells that adhere to each other to produce a colony of microorganisms, which is additionally adherent to a surface that may be living or nonliving [1]. The cells within the biofilm secrete an extracellular polymeric substance (EPS) that encases and protects this colony from host responses and potential drug treatments [2]. Biofilms occur on a wide range of artificial and natural surfaces. Biofilm formation has been found in a variety of anatomic settings including wounds, the ear, and lungs; it accounts for more than 80% of human microbial infections [3].

In some cases, altered pathophysiology may provide a favorable setting for biofilm formation, such as the altered mucus composition in patients with cystic fibrosis (CF). Mucin, the glycoprotein responsible for viscoelastic properties of mucus, is overproduced, and abnormal glycosylation patterns are observed within CF patients [4]. The mucus-filled environment gives rise to a breeding ground of bacterial development. Chronic infection via *Pseudomonas aeruginosa*, a Gram-negative bacterium notorious for its antibiotic resistance due to biofilm formation, is common within 80% of CF patients [3]. Medical devices and instruments may also be contaminated with *P. aeruginosa*; thus, hospital-acquired infections are not uncommon [5]. Patients infected with *P. aeruginosa* are given antibiotic treatments, such as tobramycin and colistin, that are only effective in high doses to treat biofilms. These high dosages, in turn, induce systemic toxicities [6], and their prolonged use can lead to antibiotic resistance [7].

Pharmacokinetic–pharmacodynamic (PK-PD) models are frequently used as tools to design dosing and administration protocols and as frameworks to interpret experimental results in preclinical studies. For antibiotic treatments of infection, they are often implemented using static parameters, such as minimum inhibitory concentration (MIC) for the pharmacodynamics and maximum drug concentration (C_max_) or drug area under the curve (AUC) for the pharmacokinetics [8]. The physical barriers posed and community nature of a bacterial biofilm are such that it may be necessary to incorporate additional factors, such as the dynamics of drug transport and the delayed, cooperative effects of drugs on biofilm bacteria, in order to better describe drug response. The better experimental quantification of the dynamics of biofilm response to various drug treatments and their incorporation into pharmacodynamic (PD) models are crucial in understanding and incorporating the concentration-dependent and dynamic effects involved in overcoming biofilm infections. For example, recent developments in the application of confocal laser scanning microscopy with flow chambers has enabled the monitoring of the real-time killing of bacterial biofilms [9,10,11].

In the present work, a rich dataset was used to validate a novel PD model for the killing of *P. aeruginosa* in biofilms via tobramycin and colistin. The proposed model incorporates three essential components: drug diffusion to the biofilm, nonlinear drug concentration effects on cellular damage, and a passage through multiple transit states by which the cells eventually become nonviable. This model was applied to various drug administration experiments that reflect the dynamic nature of biofilm as well as the cellular mechanisms involved in response to the drug. Specifically, the model was fit to experiments in which *P. aeruginosa* biofilms received either transient or continuous exposure to one drug or a combination of two drugs.

## 2. Materials and Methods

### 2.1. Experimental Dataset

A pharmacodynamic model, which captures the effects of drug concentration, drug diffusion, and cell transit through several states, ultimately leading to cell death, was developed to describe previously reported data regarding tobramycin and colistin treatment of *Pseudomonas aeruginosa* (GFP-tagged strain PA14) biofilms cultured in a well-defined flow cell at 30 °C in minimal medium with citrate (0.5 mM) at 3.3 mL/h [9]. In the experiments, biofilm populations were established for 48 h under flow. Subsequently, the biofilms were provided continuous or transient treatments of drugs using the flow cell system, and data were collected continuously for up to 24 h. The transit time of the drug within the tube was approximately 90 min, which was accounted for in our model by subtracting 1.5 h from the raw data. Propidium iodide (PI) dye was included in the flow solution to stain the nonviable biomass, and the resulting fluorescence was recorded via automated microscopy and normalized to the maximum fluorescence intensity recorded. As a result, the experimentally reported quantity to which model predictions were compared was the “Relative Biovolume”, representing the normalized values of dead biovolume. A negative control of medium only and a positive control of 70% ethanol served as checks on the fluorescence intensity data.

### 2.2. Mathematical Model

In the proposed pharmacodynamic model (Figure 1), exposure to drugs induces healthy biofilm cells (B) to enter and progress through one or more transit states (D_1_, D_2_, …) in which the cell membrane integrity is maintained (i.e., they do not stain with propidium iodide) but the cells are no longer able to divide. Progression from the last transit state produces dead cells (X), corresponding experimentally to the nonviable biovolume. Mass balances were used to derive kinetic equations describing the populations of healthy biofilm cells, the respective transit compartments, and dead cells. For tobramycin administration, the number of transit compartments was determined via optimization to be five, leading to the following set of balance equations:(1)$$\frac{dB^{*}}{dt}=B^{*}\cdot \left[\mu \cdot \left(1-B^{*}-D_{1}^{*}-D_{2}^{*}-D_{3}^{*}-D_{4}^{*}-D_{5}^{*}-X^{*}\right)-k_{s}\left(\alpha ,\beta \right)C_{0}^{\gamma }\right],$$
(2)$$\frac{dD_{1}^{*}}{dt}=k_{s}\left(\alpha ,\beta \right)C_{0}^{\gamma }\cdot B^{*}-k_{t}\cdot D_{1}^{*},$$
(3)$$\frac{dD_{2}^{*}}{dt}=k_{t}\cdot \left(D_{1}^{*}-D_{2}^{*}\right),$$
(4)$$\frac{dD_{3}^{*}}{dt}=k_{t}\cdot \left(D_{2}^{*}-D_{3}^{*}\right),$$
(5)$$\frac{dD_{4}^{*}}{dt}=k_{t}\cdot \left(D_{3}^{*}-D_{4}^{*}\right),$$
(6)$$\frac{dD_{5}^{*}}{dt}=k_{t}\cdot \left(D_{4}^{*}-D_{5}^{*}\right),$$
(7)$$\frac{dX^{*}}{dt}=k_{t}\cdot D_{5}^{*}.\;$$

Colistin administration followed the same model structure; however, there was only one transit compartment as opposed to five.

In the above equations, the values for each compartment were normalized, as indicated by the asterisks, to the maximum biovolume observed, in accordance with the experimental data [9]. It is assumed that all of the cells start in the healthy biofilm state, from which they can proliferate with a specific growth rate $\left(\mu \right)$ that is modified with a capacity constraint term (Equation (1)). The rate of healthy cell entry into the transit rates is given in terms of a rate constant, *k*_s_, and the bulk concentration, *C*_0_, raised to a cooperativity factor, *γ* (Equation (2)). The rate constant is proportional to the diffusive flux (Appendix A) and can be expressed in terms of two model parameters, *α* and *β*, each of which is a grouping of physical constants, to give:(8)$$k_{s}=\alpha \left[1+2\overset{\infty }{\underset{n=1}{\sum }}{\left(-1\right)}^{n}e^{-n^{2}\beta t}\right].\;$$

Biofilm cells affected by tobramycin eventually progress through five compartmental transit states (Equations (2)–(6)), at a rate of $k_{t}$ per state, in which they become progressively less viable than the previous state. In the final compartment $\left(X^{*}\right)$, the biofilm cells are nonviable (dead). It is this quantity that can be compared with the measured nonviable biovolume.

The coupled set of ordinary differential Equations (1)–(7) was solved using an ode45 solver in MATLAB R2020b, where the initial relative density of the biofilm state was set to 0.81 for the tobramycin treatment and 0.89 for the colistin treatment, and the rest of the compartments started with no biovolume. For each drug (tobramycin and colistin), five adjustable model parameters ($\mu ,\;f_{c},$ $k_{t},$ α, and β) were fit to the composite experimental data [9] across varying respective concentrations and time courses of 24 h. An error function was first created to evaluate the squared difference between the output of the model for a given set of parameters and the given data at a specific timepoint. This function was then minimized using the MATLAB implementation of the genetic algorithm (ga), which produced the desired parameter values. The initial condition was essentially an extra parameter within the model. To find these values for each respective drug, values ranging between 0.70 and 0.95 were tested, and the errors from the data and model output were compared. The initial concentration producing the least errors was then used.

## 3. Results

A detailed pharmacodynamic model was proposed to describe the dynamic response of *Pseudomonas aeruginosa* to the antibiotics, tobramycin and colistin (Figure 1). In this model, the drug first must diffuse through a boundary layer to get to the biofilm. When the drug reaches the biofilm, the biofilm cells go through a progression of transit compartments, the number of which is specific to and reflects the mechanism of action of the drug. Progression through the transit compartments is irreversible; consequently, the cells ultimately die after passing through them. The model was fit to continuous-time data for the killing of *P. aeruginosa* in a flow-cell chamber [9]. The available dataset used to validate this model consists of the amount of dead biofilm (relative biovolume) as a function of time for several different drug concentrations for two different treatment protocols (tobramycin and colistin).

The model was first tested on data for *P. aeruginosa* treated with tobramycin (TOB). The mechanism of action for tobramycin involves binding to the 30S ribosomal unit, thereby inhibiting protein synthesis, which gradually incapacitates the bacterium and ultimately induces cell death [12]. This is a prolonged process, which was modeled using five transit compartments, as described in Section 2. At the TOB 20 µg/mL concentration, experimentally, there is a delay of approximately 5.5 h between drug exposure (with the dead volume of the system already taken into account) and the emergence of nonviable biovolume, which subsequently increases rapidly (Figure 2). This behavior is captured by the model following a continuous treatment of TOB for 24 h. The time required for the drug to diffuse to the biofilm is seen within the flat region of the graph, and as the cells progress through the transit compartments, they are still viable until death in the X* compartment. After the composite time for drug diffusion and cellular compartment transit, there is a rapid increase in the number of dead cells observed experimentally and predicted by the model. For the other studied concentrations of tobramycin, 5 and 50 µg/mL, the proposed model shows the same pattern of delay, progression through the transit compartments, and increase in the dead biovolume population, all of which are consistent with the experimental results [9].

A useful pharmacodynamic model should be able to capture not only the dynamics, but also the concentration (dose) dependence of response. To this end, the model’s parameters were fit to the ensemble data of 5, 20, and 50 µg/mL TOB exposure to produce one set of fit parameters (Table 1). This single set of parameters successfully describes the dynamics of cell killing for the concentration range of 5–50 µg/mL (Figure 3). The dependence of the lag time before the onset of dead biovolume is consistent with the drug diffusion aspect of the model. Delay due to diffusion is seen for all drug concentrations, and it is amplified for lower concentrations. Because the diffusive flux of drug to the biofilm cells is directly proportional to the concentration driving force, less delay and higher dead biovolume concentrations are observed at shorter times for greater drug concentrations. Consequently, TOB concentrations of 20 and 50 µg/mL yield shorter lag times as compared to 5 µg/mL (Figure 3).

A key application of a pharmacodynamic model is its use to predict the response to varying drug administration protocols. Experimental data are available for the response of *P. aeruginosa* to transient exposure to TOB, where the drug is administered for the first four hours and then turned off for the remaining twenty hours. The pharmacodynamic model predicts that drug effects will continue to be observed after the removal of a drug from the bulk, due to the continued flux of the remaining drug through the boundary layer and the continued progression of cells through the transit compartments. As a result, a sharp increase in dead cell biovolume is predicted by the model and observed experimentally during the period from 5 to 20 h after initial exposure, i.e., after the drug is turned off (Figure 4). At the higher concentrations of 20 and 50 µg/mL, regrowth is observed in the experimental model about 12 h after the drug administration ceases. Only at the lowest TOB concentration of 5 µg/mL is there a reduction in killing in the transient exposure experiment as compared to continuous exposure. This behavior is explained by the model as being due to an insufficient amount of drug having diffused into the boundary layer during the four hours of drug exposure (Figure 4).

Experimental data on the treatment of *P. aeruginosa* in the same flow system are available for colistin, whose mechanism of action provides a contrast to that of tobramycin. Colistin is a lipopeptide which binds to phospholipids found on the membrane of the cells and replaces cations [13]. This induces cell rupture and leakage of the inner contents of the cell, leading to death. Because this drug has a more rapid mode of killing than tobramycin, the pharmacodynamic model was modified to contain only one transit compartment (Figure 5A), such that the progression of cells from the exposure to the drug to cell death is more rapid than for tobramycin. Analogously to tobramycin, we fit data for multiple concentrations of colistin (CST) into one set of parameters and used these to model various concentrations of CST administered continuously over a period of twelve hours.

As seen with TOB, a delay is observed in the response to CST due to the time required for diffusion through the boundary layer (Figure 5B). However, the delay is shorter due to the existence of only one transit compartment and more rapid transit throughout. As with TOB, a single set of parameters accurately describes the ensemble of data over the tested concentration range (Figure 6). These same parameters were used when applying the model to a transient exposure to CST, where the drug was administered for the first four hours and shut off for the remaining time (Figure 7). The same trend is seen as in the continuous treatment, where at higher concentrations, the diffusive flux of CST is greater, therefore resulting in less delay and the rapid onset of cell killing. Additionally, at these high concentrations of colistin, all of the biofilm cells are observed to become nonviable at earlier time points in comparison to TOB, again largely due to drug-treated cells spending less time in transit compartments.

Since TOB and CST have different mechanisms of action, they might produce additive or synergistic effects when used in combination. If there are no strong synergies or antagonisms, the original model may be able to predict outcomes of combined treatments using only the parameters determined earlier for each respective drug. The model proposed for this mechanism involves a combination of both treatments running in parallel (Figure 8A). It proved necessary to add a path by which the biofilm cells could initially be affected by TOB or CST, and the cells in transit due to (slower-acting) TOB exposure could be killed directly by (faster-acting) CST. It is assumed that cells in a transit compartment due to TOB were equally likely as naïve cells to be affected by CST; thus, this path does not introduce any additional fitting parameters into the model. Because of the more rapid killing mechanism of CST, the response to CST dominates the experimentally observed and model behaviors (Figure 8B), where the biofilm cell death occurs at earlier times, even with lower concentrations of CST.

## 4. Discussion

Bacterial biofilms are a significant problem in human infections because they form communities that both pose physical barriers to drug transport and allow metabolic adaptations that can alter the pharmacology of antibiotic treatment [14,15]. An improved understanding of the response of biofilm-associated bacteria to antibiotic treatment is needed to optimize the administration route and timing of existing drugs and to focus efforts on novel antibiotic development. Experimental datasets wherein the response of a biofilm to treatment is monitored continuously over time provide a signature of the pharmacologic response. The development of a mathematical model that captures this response serves as a complementary tool that enables the interpretation of these data in terms of physicochemical mechanisms.

Conventional pharmacologic expressions based on receptor theory are used to describe the pharmacodynamics (PD) of bacterial response to antibiotics [16,17,18]. These in turn are incorporated into pharmacokinetic–pharmacodynamic (PK-PD) models, which are an important tool in understanding the dose and time dependence of outcomes in preclinical studies and serve as the basis for early-phase clinical dose and administration scheduling [19,20]. Traditionally, the dynamics in PK-PD models of anti-infectives are dictated by the distribution of the drug, and the concentration dependence is reflected in the pharmacodynamic expression. The simplest such expression, which is commonly employed in practice, treats the encounter between the drug and target cell as a first-order reaction resulting in instantaneous cell killing [21]. This approach does not capture important trends observed in preclinical and human infections, including a delay between drug exposure and drug effect and more complex dose–response relationships.

More elaborate mathematical models have been proposed to describe the growth and treatment of biofilms, taking into account physical effects such as the diffusion of substrate for growth and the diffusion of antibiotic for killing within the biofilm, the interfacial detachment of biofilm-associated cells, advection, and chemotaxis [22,23,24,25,26]. Furthermore, additional cellular states that reflect heterogeneity of response, e.g., persister states [27], or transitional states due to cell damage [28], have been incorporated. These models promote our quantitative understanding of the role that these physical and cell physiologic effects can play in the growth and antibiotic treatment of biofilms. However, the incorporation of mechanisms that depend on both space and time involves partial differential equation-based continuous or agent-based simulation models that do not incorporate readily into PK-PD models [29,30]. We sought to develop a model with sufficient mechanistic detail so as to describe the dynamics of the biofilm response while still being tractable for eventual incorporation into a PK-PD framework.

Recent experiments that monitor biofilm response dynamically demonstrate that there is a delay between the onset of drug exposure and cell killing and that the magnitude of the delay depends on the particular drug being used [9,21]. Thus, while diffusion can play a role in the temporal response, cell physiology and the therapeutic mechanism of action are also evidently important. Based on these observations, we developed a pharmacodynamic model whose response has two critical aspects: the diffusion of a drug through a boundary layer to the cells, and a cell physiological response in which a cascade of events is initiated whose number and rates can be modified depending on the mechanism of action of the drug. Specifically, we introduced “transit compartments” to account for cell states that are affected by drugs: nonproliferative, but not yet dead (Figure 1). Incorporating these elements, the model has five adjustable parameters with distinct mechanistic interpretations.

The parameter *µ_B_* represents the specific growth rate of the biofilm. As little cell growth is observed during the time course of the experiments being modeled, its value is low, and no finer detail needs to be incorporated. The inclusion of drug diffusion results in two lumped parameters, α and β (Equation (6) and Appendix A). The β parameter is the value of $\frac{\pi D}{H^{2}}$, where *D* is the diffusion coefficient, and *H* is the thickness of the diffusion layer, which is a combination of the hydrodynamic layer resulting from the experimental setup in a flow cell, as well as the physical barrier imposed by the biofilm itself. The lumped parameter *β* results from the scaling of the diffusion problem and is the inverse of the characteristic time for diffusion. Using the biofilm thickness of ~20 μm, the fit values of *β* would correspond to diffusion coefficients (1.3–5.2 × 10^−5^ mm^2^/h). These values are several orders of magnitude lower than those for typical drugs in water [31], suggesting that the diffusivity of the drugs is reduced in the biofilm and/or there is also a mass transfer boundary layer [32]. For this reason, *β* was retained as a fit, rather than fixed, parameter. Since the boundary layer thickness should be the same for both drugs, the slightly higher value fit for CST than for TOB can be interpreted as a higher effective diffusion coefficient for the former compound. Although CST has a higher molecular weight than TOB, it has biosurfactant properties that may allow it to diffuse (penetrate) more rapidly in the biofilm barrier [33].

The α value is a lumping of $\frac{k_{C}D}{H}$, which includes the aforementioned parameters that describe diffusive flux, as well as a rate constant, $k_{C}$, to denote the rate at which biofilm cells are affected by the drug and enter into a transit compartment to begin its death cascade. The γ value is a purely pharmacodynamic parameter representing cooperativity in terms of the drug binding to and poisoning of the biofilm cells. The last parameter, *k_t_*, corresponds to the intercompartmental transit rate of the drug. This is not typically found in other PD models that are designed to capture data at one time point; however, it is critical in capturing the overall dynamic behavior of the drug and its effect on biofilm killing.

In order to better understand the influence of model structure and parameter values on the model output, several analyses were performed. First, each component of the model was removed in turn and the model re-fit to the experimental data (Figure 9), with the exception of the cell killing component (*α* parameter), which would give a trivial result. The removal of either the diffusion term or the transit compartment terms significantly reduced the ability of the model to capture the dynamics of the biofilm response across the three different concentration levels of TOB. Likewise, removing cooperativity from the model (i.e., constraining the cooperativity parameter, *γ*, to a value of one) prevented an accurate reflection of the concentration dependence. The removal of bacterial growth, *μ*, did not have a major effect on the model output. However, when neglecting this parameter in the optimization program, the error between the model and data was increased by 60% in comparison to incorporating growth rate (Table 2). As a result, the growth rate was retained in the model.

The effect of these five parameters on the model output can be further understood through a parametric sensitivity analysis in which each parameter’s value is varied while holding all others constant (Figure 10). It is evident that variation in the value of *k_t_* has the greatest impact on the model overall. Conceptually, this is expected as the intercompartmental transit rate dictates progress through the “death cascade” as well as the cellular response dynamics to the drug. The value of *β* dictates the delay due to diffusion between drug administration and cellular effects. It couples with *k_t_* and has a strong influence on the output. The pharmacologic rate constant, *α*, has a modest effect on the output, while the cooperativity, *γ*, exerts a stronger influence.

The number of transit compartments can be considered as an additional model parameter. We varied the number systematically, refitting the model each time to determine the value most consistent with the experimental data (Figure 11). For CST treatment, it was found that one compartment produces the most accurate model, whereas for TOB, five compartments provides the best fit. For either TOB or CST treatment, as the number of compartments increases, *k_t_* increases in order to mimic the effect of one transit compartment (Appendix A).

The proposed model fit with a single set of parameters for each respective drug was able to reproduce the response to drug concentrations that vary over an order of magnitude in both continuous and transient combined exposure (Figure 3, Figure 4, Figure 6 and Figure 7). It is challenging for a model to capture both concentration (dose) and time effects. The ability of the present model to do so with a single set of fit parameters is promising. Furthermore, the effect of combined treatment was captured accurately using parameters for individual drug treatments (Figure 8B).

Areas where agreement was less robust point to limitations in the model and/or in the experimental dataset. For instance, in the transient exposure experiments (Figure 4), a reduction in biovolume (adherent, dead cells) occurred around 20 h after treatment onset. This represents detachment, which has been incorporated into some biophysical models of biofilm growth [34]. However, it was not considered in the present work, as the physical location of dead cells (adherent versus detached) is not of great interest in pharmacologic applications. In addition, the model somewhat underpredicts the extent of cell death throughout the transient experiment. This underprediction is because the model parameters were fit based on the continuous treatment experiments (Figure 3), and the rate of cell killing was more rapid during transient exposure, a result that most likely reflects experimental variation rather than a physical or pharmacologic effect. Another limitation is that, because experimental data were collected for only 24 h, pharmacologic effects occurring at longer times might not be captured accurately in the model. This would include effects such as regrowth of the biofilm and the development of drug resistance, both of which tend to evolve over longer periods of time. It should be noted that the experimental data were obtained on mature biofilms (grown for 48 h before treatment), which almost completely covered the surface of the flow cell substrate [9]. Under these conditions, it is expected that the biofilm growth rate is low, and it is likely that heterogeneities in biofilm structure would be less important than in a developing biofilm. These differences are of interest in the development of pharmacodynamic indices for PK-PD models [35,36].

## 5. Conclusions

In summary, we have shown that a pharmacodynamic model which integrates the diffusion of a drug from the bulk to the cells, drug–cell interactions, and a series of transit compartments for affected cells is able to accurately describe the dynamics of *Pseudomonas aeruginosa’s* response to tobramycin, colistin, and their combinations. The model was fit to an ensemble of data covering multiple drug concentrations to obtain one set of fit parameters for each drug. Among these, the specific growth rate proved inconsequential during the time course of the experiments studied, but each of the other four parameters exerted a distinct influence on the model output and contributed to its ability to capture experimentally observed dynamics. Overall, the model is robust enough to show the general behaviors of TOB, CST, and the combination of the two drugs at various concentrations. The model allows the interpretation of the slower-acting TOB as resulting from a mechanism involving multiple cellular steps (“transit states”) and suggests that more sustained treatment is necessary to eradicate biofilms with this drug. This pharmacodynamic model can be paired with a pharmacokinetic description in vivo to predict a drug’s effect on an infection. This could be of potential interest in tissues such as lung where both systemic and regional (e.g., pulmonary) delivery are possible [37]. Our model can be useful in simulating effects of different strategies in drug administration and scheduling to promote the better eradication of challenging biofilm infections.

## Figures and Tables

**Figure 1 biomedicines-11-02316-f001:**
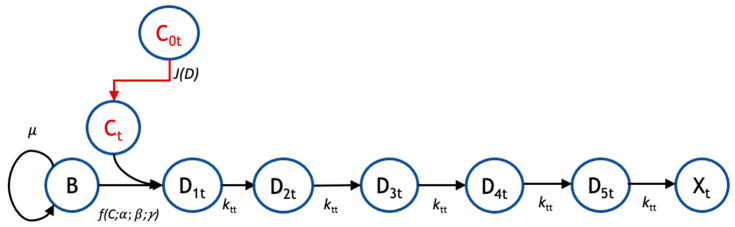
Pharmacodynamic model structure for tobramycin. The pharmacodynamic model for response to tobramycin (subscript ‘t’) tracks the transit of biofilm cells going from a viable (B) to nonviable state (X_t_) following administration of tobramycin at bulk concentration *C*_0t_. There is a flux, *J*(*D*) of drug from the bulk to the biofilm cells, where the local concentration is *C*_t_. For tobramycin, there are five transit compartments (D_1t_, D_2t_, D_3t_, D_4t_, and D_5t_) mediating the cellular response to drug. Growth is governed by a specific growth rate, *μ*; the coupled diffusion and pharmacodynamic response are subject to parameters *α*, *β*, and *γ*; and the transit rate to subsequent compartments is given by *k*_tt_.

**Figure 2 biomedicines-11-02316-f002:**
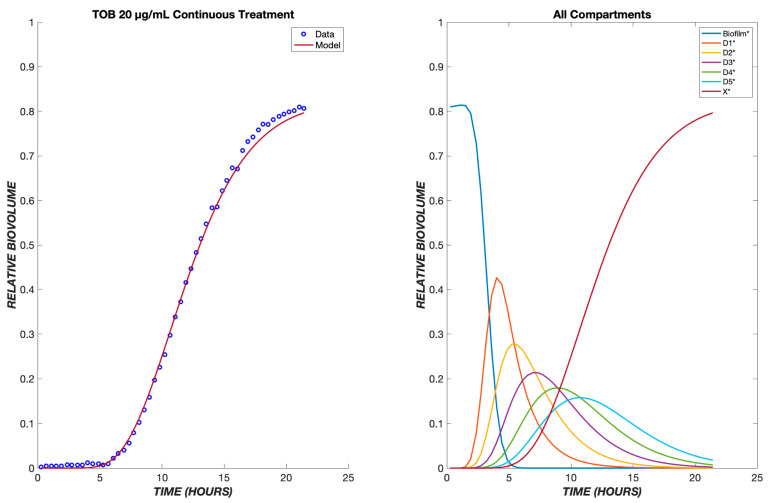
Dynamics of *Pseudomonas aeruginosa* killing in response to tobramycin at 20 µg/mL. The left panel shows the model comparison of the dead cells, shown in red, with the experimental data, shown in blue. The right panel shows the populations in each of the cellular compartments over the same time course.

**Figure 3 biomedicines-11-02316-f003:**
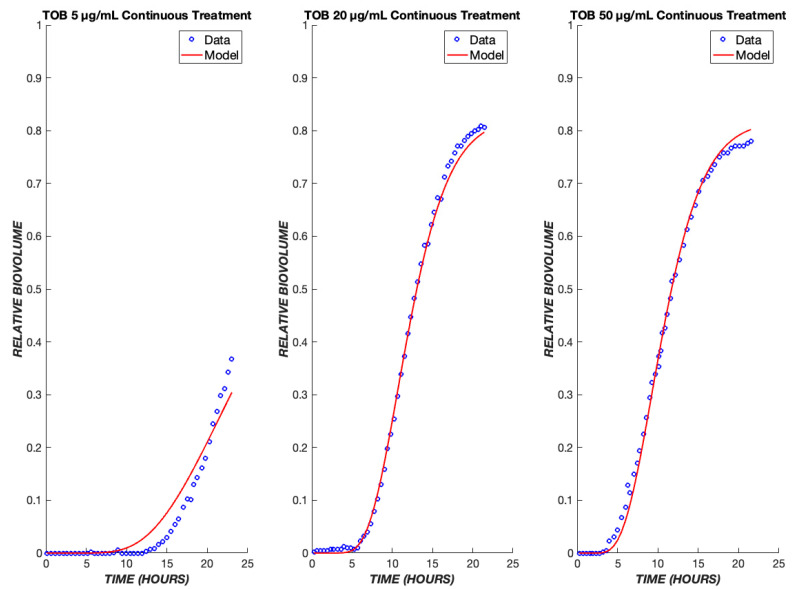
Tobramycin model fits across a ten-fold concentration range. The PD model was used to simulate the treatment of biofilms treated with tobramycin at drug concentrations of 5, 20, and 50 µg/mL. The model was fit to the ensemble data of all three concentrations. The experimental data are shown in blue, and the model is shown in red.

**Figure 4 biomedicines-11-02316-f004:**
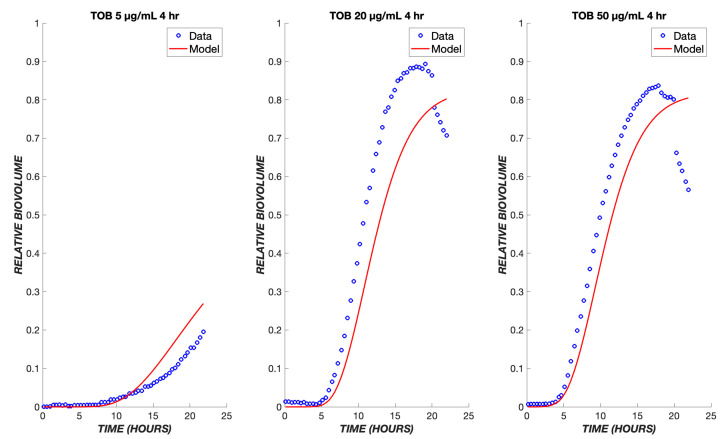
Transient exposure of biofilms to tobramycin. The same set of parameters for the continuous data was used to simulate the treatment of biofilms using tobramycin transiently for four hours at the same drug concentrations of 5, 20, and 50 µg/mL. The experimental data are shown in blue, and the model is shown in red.

**Figure 5 biomedicines-11-02316-f005:**
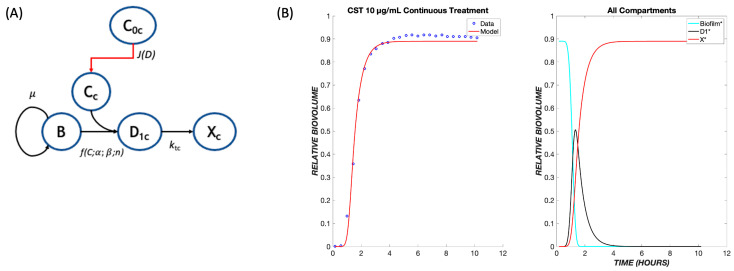
The pharmacodynamic model for colistin. (**A**) Model structure with a single transit compartment. The symbols have the same meanings as in Figure 1, with subscripts changed from t (tobramycin) to c (colistin). (**B**) Data and model fit for colistin killing of *P. aeruginosa* biofilms at 10 µg/mL. The model fit uses parameters determined by fitting to the ensemble of colistin continuous response data.

**Figure 6 biomedicines-11-02316-f006:**
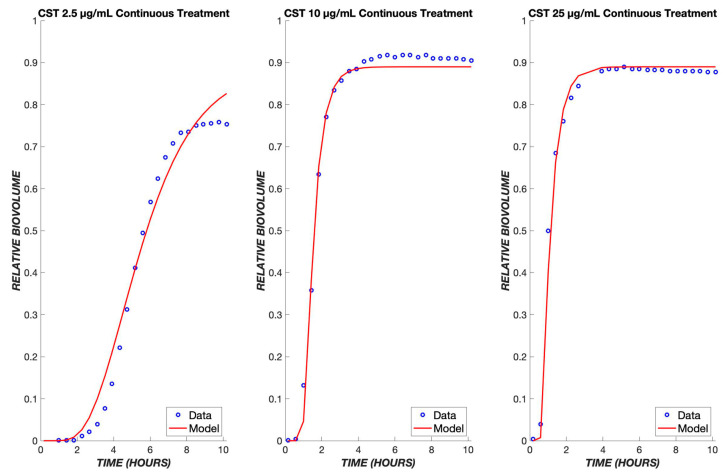
Colistin model fits across a ten-fold concentration range. The PD model was used to simulate the treatment of biofilms using colistin concentrations of 2.5, 10, and 25 µg/mL. The data of all three concentrations were used to fit one set of parameters, which was used to model the various concentrations shown above. The experimental data are shown in blue, and the model is shown in red.

**Figure 7 biomedicines-11-02316-f007:**
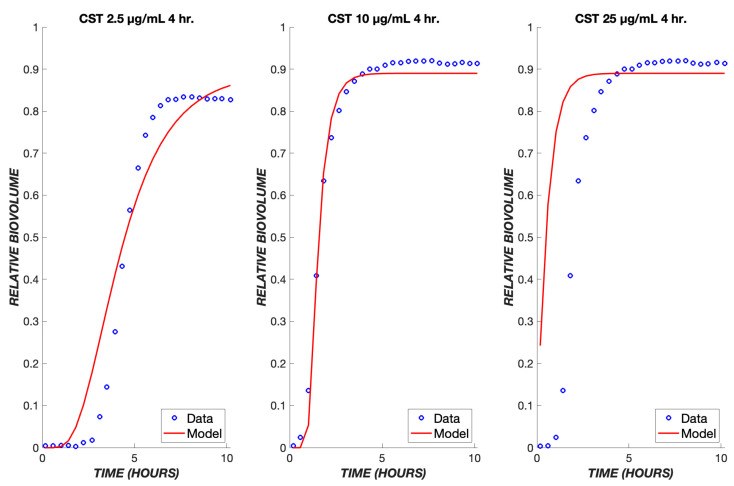
Transient exposure of biofilms to colistin. The same set of parameters for the continuous data was used to simulate the treatment of biofilms using colistin for four hours at concentrations of 2.5, 10, and 25 µg/mL. The experimental data are shown in blue, and the model is shown in red.

**Figure 8 biomedicines-11-02316-f008:**
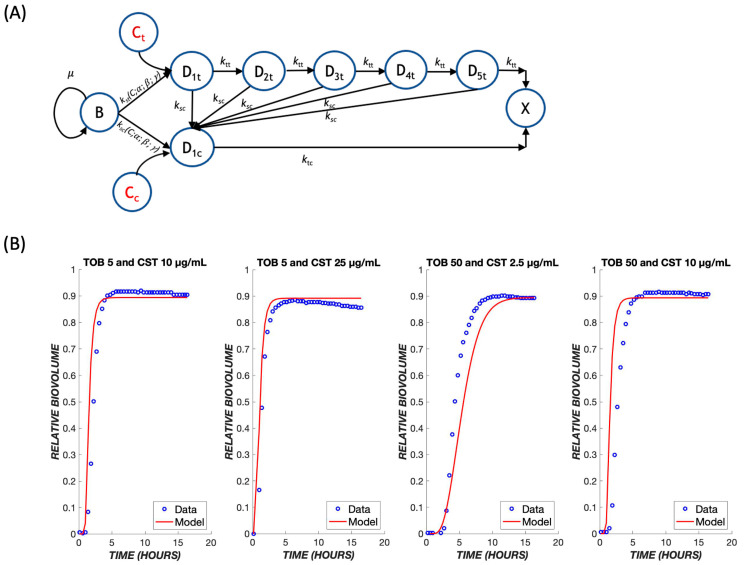
Treatment with drug combinations. (**A**) A schematic model for the combination treatment with two drugs. Transit compartments resulting from exposure to tobramycin and colistin are considered as parallel death pathways, with the possibility for crossover from the slow-acting tobramycin to fast-acting colistin pathway. The same parameters that were derived earlier for each respective drug were used to predict outcomes of combination treatments. (**B**) Combination treatments were tested using various dose combinations of tobramycin and colistin over the course of 24 h. The mathematical model is shown in red, and the experimental data are shown in blue.

**Figure 9 biomedicines-11-02316-f009:**
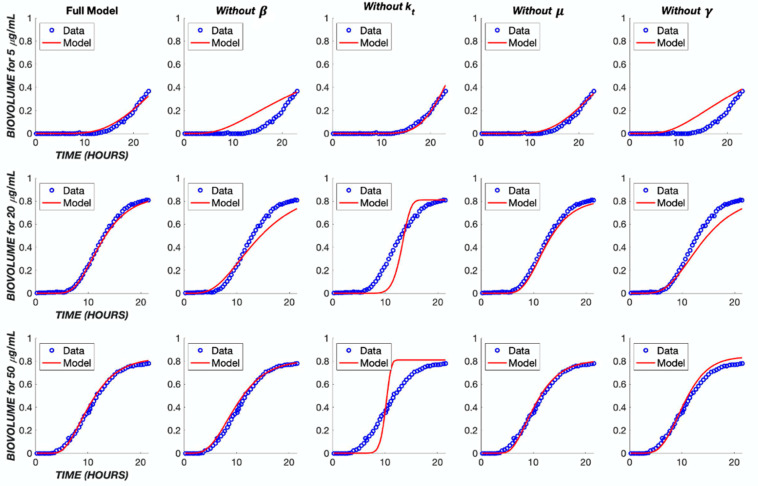
Impact of model simplifications on performance. The TOB continuous treatment was used to analyze the parameters within the model. Each TOB concentration was fit to the data without one indicated parameter. The ‘Full Model’ column shows the fit with the original parameter values found in Table 1.

**Figure 10 biomedicines-11-02316-f010:**
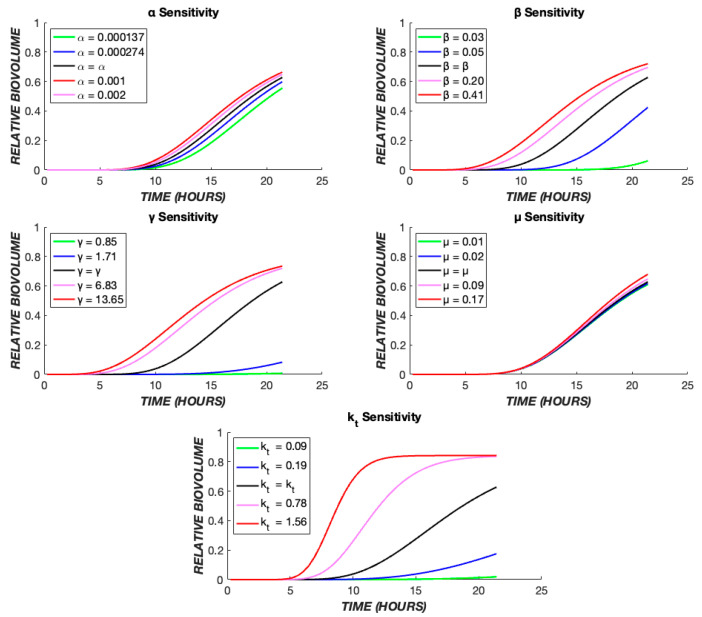
Parametric sensitivity analysis. The TOB 20 µg/mL continuous data were used as the base case. The time course of dead biovolume (X*) production was compared for values of each parameter, varied one at a time. The black line represents the output when the parameter is at its best fit value (Table 1), the red and pink lines show the effect of higher values, and the green and blue lines illustrate the effect of lower values of each parameter.

**Figure 11 biomedicines-11-02316-f011:**
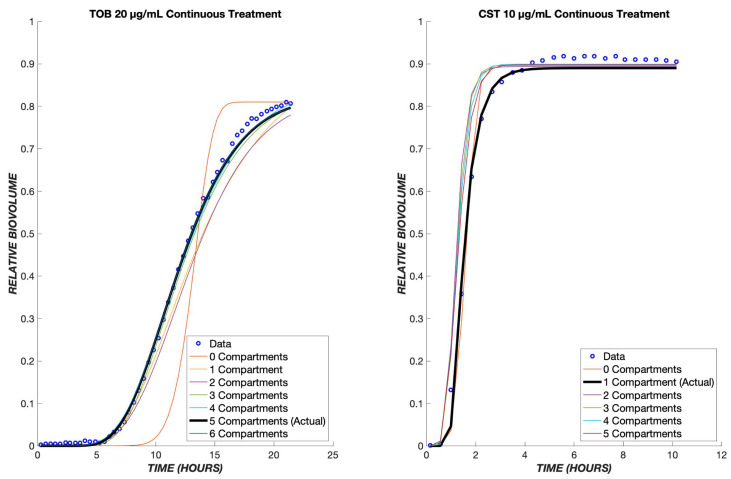
Optimal number of transit compartments. The TOB 20 µg/mL and CST 10 µg/mL continuous data were used to determine the ideal number of compartments for the model. On the left graph, the model with three compartments gave the least error between the model and data for tobramycin. The graph on the right shows the model with a differing number of compartments for colistin, and as shown, only one compartment produced the least error between the model and data. The parameter values for each compartment can be found in the tables below.

**Table 1 biomedicines-11-02316-t001:** Pharmacodynamic model parameters.

Parameter	Description	Units	TOB Value	CST Value
*μ* _B_	Growth rate of biofilm population	h^−1^	0.0321	0.0001
*α*	Rate constant for drug effect on biofilm	(μg/mL)^γ−1^ h^−1^	0.0002	0.0082
*β*	Normalized drug diffusivity	h^−1^	0.2088	0.3986
*γ*	Cooperativity in drug effect on biofilm		3.5330	4.4313
*k* _t_	Intercompartmental transit rate of drug	h^−1^	0.5424	1.8924

**Table 2 biomedicines-11-02316-t002:** Impact of removing model components on total error.

Parameter	Full Model	Without *μ*_B_	Without *β*	Without *γ*	Without *k*_t_
*μ* _B_	0.0321	-	0.5372	0.0289	0.0000
*α*	0.0002	0.0012	0.0072	0.0094	0.0351
*β*	0.2088	0.0815	-	0.2480	0.0200
*γ*	3.5330	3.2862	1.0705	-	3.4315
*k* _t_	0.5424	0.4370	0.4228	0.8053	-
Error	0.0561	0.0971	0.4337	0.7883	1.6449

## Data Availability

The digitized experimental data, as well as the Matlab codes used for mathematical model solution and figure generation, are available in a GitHub repository at https://github.com/RothLabRU/Biofilm-PD.

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
