# Peer review of "Pharmacodynamic Model of the Dynamic Response of Pseudomonas aeruginosa Biofilms to Antibacterial Treatments"

_biomedicines, 2023, doi:10.3390/biomedicines11082316_

Round 1
Reviewer 1 Report
In Title, instead of drug treatments, Antibacterial treatments is better.
Introduction part:
1. Authors used a novel PD model to examine the killing of P. aeruginosa in biofilm by tobramycin and colistin. Why did the authors select two antibiotics?
Materials and Methods part:
1. It is well-organized with good references. But I want the authors to provide a dioramic figure of this PD model. If you use flow-through cell, which kind of media did you use? Mention the flow rate and temperature you have used in the system.
2. In lines 149,153, 200 authors used tobramycin (TOB) but so many time again used tobramycin in full form. So on paper its better to use TOB instead of tobramycin. And the same thing happens in the case of colistin in line 217, 393, and table 1.
Results part:
1. Authors used three concentrations of tobramycin and colistin for the experiments. Why didn't the authors use positive and negative control drug such as antibiofilm inhibitors?
2. Quorum sensing is closely related to the biofilm formation of bacteria. Do the used two antibiotics have quorum-sensing formation? Authors need to explain the relationship between antibacterial activity and quorum sensing activity.
Conclusion part:
1. Authors told, "Our model can be useful in simulating effects of different strategies in drug administration and scheduling to promote better eradication of challenging biofilm infections". With your results in the present result, can you suggest an interval or dose in drug scheduling eradication?
Reviewer 2 Report
The authors developed a compartmental model structure in which the key features are diffusion of drug through a boundary layer to the bacteria, concentration dependent interactions with bacteria, and passage of the bacteria through successive transit states before death. As a result, this PD model is integrated with PK descriptions to describe in vivo antibiotic response dynamics and to predict drug delivery strategies for improved control of bacterial lung infections.
<comments>
1. This paper is generally well written. Although the results of this study contain the valuable results, they are merely database analyzes, and no clear recommendations have been made as to what aspects of the results are novel or how to develop these results. Therefore, the scientific value is unknown.
2.The description of the paper should be revised so that non-expert readers can understand the analysis method and its interpretations. Since the leaders of this journal are not necessarily experts who can understand the results of this study, the analysis methods and results considerations.
3. There is a lack of information on the Pseudomonas aeruginosa strains and drug susceptibility assumed in this study. If it is intended for practical clinical application, this information is necessary.
Round 2
Reviewer 2 Report
I don't see any particular points to be pointed out.